6mA-Pred: identifying DNA N6-methyladenine sites based on deep learning

Huang Qianfei 1
Zhou Wenyang 2
Guo Fei 1
Xu Lei 3 csleixu@szpt.edu.cn
Zhang Lichao 4 lczhang5354@szu.edu.cn
1 College of Intelligence and Computing, Tianjin University , Tianjin , China
2 School of Life Science and Technology, Harbin Institute of Technology , Harbin , China
3 School of Electronic and Communication Engineering, Shenzhen Polytechnic , Shenzhen , China
4 School of Intelligent Manufacturing and Equipment, Shenzhen Institute of Information Technology , Shenzhen , China
Gillespie Joseph
Electronic publication date: 2021 Feb 3
Publication date: 2021
Volume: 9
Electronic Location ID: e10813
Received 2020 Oct 28; Accepted 2020 Dec 30
Copyright: © 2021 Huang et al.
Copyright year: 2021
Copyright holder: Huang et al.
License: This is an open access article distributed under the terms of the Creative Commons Attribution License, which permits unrestricted use, distribution, reproduction and adaptation in any medium and for any purpose provided that it is properly attributed. For attribution, the original author(s), title, publication source (PeerJ) and either DOI or URL of the article must be cited.
License URL: https://creativecommons.org/licenses/by/4.0/

Keywords: 6mA, LSTM, Attention

Funding: Natural Science Foundation of China 61902259 Natural Science Foundation of Guangdong province 2018A0303130084 This work was supported by the Natural Science Foundation of China (No. 61902259) and the Natural Science Foundation of Guangdong province (grant no. 2018A0303130084). The funders had no role in study design, data collection and analysis, decision to publish, or preparation of the manuscript.

==============================
With the accumulation of data on 6mA modification sites, an increasing number of scholars have begun to focus on the identification of 6mA sites. Despite the recognized importance of 6mA sites, methods for their identification remain lacking, with most existing methods being aimed at their identification in individual species. In the present study, we aimed to develop an identification method suitable for multiple species. Based on previous research, we propose a method for 6mA site recognition. Our experiments prove that the proposed 6mA-Pred method is effective for identifying 6mA sites in genes from taxa such as rice, Mus musculus, and human. A series of experimental results show that 6mA-Pred is an excellent method. We provide the source code used in the study, which can be obtained from http://39.100.246.211:5004/6mA_Pred/.

Introduction

DNA modification sites play vital roles in multiple biological processes and are attracting increasing research attention. Methylation continues to be a hot topic in epigenetics, and 5mC methylation has been extensively studied (Liu, Li & Zuo, 2019). With the advancement of sequencing technology, 6mA methylation has slowly attracted increasing attention. 6mA methylation not only affects gene expression but also regulates development in plants and animals (Xu et al., 2020a). Many diseases, including cancer, are related to 6mA methylation (Chen et al., 2019a, 2019b; Xu et al., 2019a). With the progress of 6mA methylation-related research, large amounts of data have been collected. However, effective methods for 6mA site identification are lacking.

Methods for identifying modification sites have consistently been a hot spot in bioinformatics. Many methods have been studied and have achieved good results. Although research on 4mC (He, Jia & Zou, 2019) and 5mC is mature, research on the identification of 6mA modification sites has just begun. The computational method i6mA-Pred was used to identify 6mA modification sites in the rice genome with high accuracy. Several methods for identifying 6mA loci in the rice genome have been proposed, such as MM-6mAPred, iDNA-6mA-rice (Hao et al., 2019), SDM6A (Basith et al., 2019), i6mA-DNCP (Kong & Zhang, 2019) and SNNRice6mA (Yu & Dai, 2019). In addition, methods for the identification of 6mA sites in Mus musculus and humans have gradually emerged, such as iDNA6mA-PseKNC (Feng et al., 2019), csDMA (Liu et al., 2019c), SICD6mA, and 6mA-Finder (Xu et al., 2020b). Several datasets are publicly available, and many desirable features and models have been proposed. Application of the feature algorithms NCP and one-hot, feature fusion and deep learning methods has greatly accelerated the identification of 6mA-modified sites. Among the employed algorithms, SVM and RF exhibit stable performance and perform well on some datasets (Liu, Gao & Zhang, 2019; Shen, Tang & Guo, 2019; Sun et al., 2020; Wang et al., 2020a, 2020b; Yan et al., 2020; Zhou et al., 2018, 2017). In addition, the Markov model has achieved excellent results in predicting 6mA sites in the rice genome. In the application of feature methods, most researchers use multiple feature fusion methods and analyze various features. In general, the different methods have achieved good results and provided direction for subsequent research.

In the research mentioned above, most methods have employed machine learning (Patil & Chouhan, 2019; Zou, 2019; Zou & Ma, 2019) and detailed analysis of different feature methods. There are some good models that use deep learning methods, such as SNNRice6mA and SICD6mA. SNNRice6mA employs CNN (Ren et al., 2019) to build a network that works well. SICD6mA uses GRU to achieve a good network structure and has been applied extensively to datasets of two species. In this article, through a summary of the previous research work, we found that LSTM+Attention can identify the modification sites very well, and a large number of experimental results suggest that this is a very good method.

Materials and Methods

Datasets

Much research has aimed to identify 6mA sites in rice. In reviewing research from the past 2 years, we found that the amount of data on 6mA sites is increasing. We obtained datasets for three species. The first dataset is a rice dataset obtained from 6mA-RicePred (Huang et al., 2020b). This dataset was first used in i6mA-Pred (Chen et al., 2019c) and was provided by the author (Hu et al., 2019). The second dataset is a Mus musculus dataset obtained from iDNA-PseKNC, and it has achieved good results with this dataset. The third dataset is a human dataset obtained from SICD6mA and is the largest of the three datasets. Table 1 provides a summary of each dataset. The lengths of their sequences are all the same: 41 bp. Details of these datasets are provided in their source papers. We have organized the datasets, which can be obtained from https://github.com/huangqianfei0916/6ma-rice.

Table 1 All datasets.

Dataset	Positive	Negative	Total	
Rice	154,000	154,000	308,000	
Mus musculus	1,934	1,934	3,868	
Human_Train	491,885	491,885	983,770	
Human_Test	122,971	122,971	245,942	

All three data sets use CD-HIT to remove redundancy. Sequences with the similarity above 80% were excluded by using the CD-HIT program. all negative samples were 41 bp in length and the center was A, but not being detected by the SMRT sequencing technology as of 6mA. Moreover the rice dataset collected negative samples based on the ratio of GAGG, AGG and AG motifs in the positive samples. the mouse dataset removed positive samples with modQV greater than 30.

Feature encoding and classification algorithms

One-hot encoding has been used by many researchers for sequence processing with good results (Cheng, 2019; Cheng et al., 2018a; Li et al., 2020; Liu & Li, 2019; Zou et al., 2019). One-hot encoding encodes each nucleotide separately. A disadvantage of one-hot is the lack of timing. Therefore, we used Kmer word segmentation instead of one-hot to capture the relationship between bases (Zuo et al., 2017). The role of Kmer was to help Embedding generate better word vectors. We investigated both normal word segmentation and Kmer word segmentation, and the experimental results showed that Kmer word segmentation achieved superior performance. Figure 1 shows the process of Kmer word segmentation. Our test for the selection of the k value revealed three to be the most suitable value. the experimental results are shown in Fig. 2. When k is 3, the dictionary size is 64; this is not a large parameter. In the feature extraction stage, the embedding layer is used to extract features. we chose the init method for our experiment. The effect of using init or fine-tune is almost the same, and in some cases, the init method is superior. If there is an excellent pretrained model, it is also a good choice. The quality of the features largely determines the effect of the model. Embedding is a very important module in deep learning, and word2vec is one of the best embedding methods. The encoding of features can be learned dynamically, and a method of secondary learning called finetune can be achieved in deep learning. In this paper, we use simple Init embedding and Kmer word segmentation.

Figure 1 A flow chart of the structure of 6mA-Pred.

6mA-Pred includes kmer word segmentation and attention mechanism. Among them, the attention score uses the dot product method. Optimize features through the attention mechanism.

Figure 2 Performance of 6mA-Pred evaluated via independent testing based on different k-values.

(A) The performance of different k values based on the mouse data set; (B) performance of different k values based on the rice data set; (C) performance of different k values based on the human data set.

Most methods currently employed for 6mA site recognition are machine learning methods, and most of them are only effective for a single species (Cheng, 2019; Cheng et al., 2019). In reviewing the latest research, we found that there are many similarities between the attention mechanism and the recognition of 6mA sites. Furthermore, LSTM has achieved excellent performance in dealing with sequence problems (Huang et al., 2020a). In constructing the model, we did not adopt a particularly complex structure, and the complexity and effect of the model are not directly related. After feature extraction with the embedding layer, bidirectional LSTM is used to process the sequence features (Xia et al., 2019). The sequence information obtained after LSTM processing can be used to obtain a good feature vector, and this feature is a representation of the overall sequence information. Each time step of LSTM has an output that represents the sequence information up to the current time. The LSTM algorithm can be formulated as follows: (1) it=σ(Wiixt+bii+Whiht−1+bhi)

(2) ft=σ(Wifxt+bif+Whfht−1+bhf)

(3) gt=tanh⁡(Wigxt+big+Whght−1+bhg)

(4) ot=σ(Wioxt+bio+Whoht−1+bho)

(5) ct=ft∗ct−1+it∗gt

(6) ht=ot∗tanh⁡(ct)

In general, LSTM can be used to obtain an output at each time step and obtain a feature containing the sequence information (Liu, Li & Yan, 2020). We can analyze these features to obtain our expected results. The typical approach is to average this information or take the last one and then apply the fully connected layer to obtain the result. Many scholars have added other layers after LSTM to obtain good features. However, the design of these levels of network structure varies according to the specific application scenarios and problems. 6mA-Pred applies the attention mechanism to the output of LSTM and connects the fully connected layer after the attention layer.

The attention layer is added after the LSTM, and the output of the LSTM is analyzed with attention. The inner output of the final output of LSTM and the results of the previous time step can be used to generate the corresponding attention score. then, the Softmax layer is added to the attention layer to obtain the weight. The output of LSTM and this weight are weighted to obtain the final context vector. The last layer of the network is the fully connected layer, and this layer can obtain the probability of each category. Figure 1 shows the structure of the entire network and describes the Kmer word segmentation and attention mechanism. The attention mechanism adopted by 6mA-Pred is not complicated and acts directly on the output of LSTM. The purpose of 6mA-Pred is to obtain the final feature through the difference between global information and local information. We know that the feature corresponding to the sequence containing the modification site is very different from the feature corresponding to the sequence not containing the modification site. Because of the differences, their final context vectors differ. We used the inner product method to obtain the attention score to reflect the intersection of global information and local information. The inner product is not the only option; other operations are possible. Self-attention in Transformer is also a good choice, but the network structure of the model is more complicated. Dot product can get the intersection between different sequences. 6mA-Pred uses this structure to increase the amount of local information in the final feature.

Performance evaluation

A good model evaluation standard is crucial for assessing the utility of a model. Different indicators can be used to reveal the advantages and disadvantages of a model from different perspectives. Sensitivity (Sn), specificity (Sp), accuracy (Acc), and Mathew’s correlation coefficient (MCC) are used to evaluate models in machine learning (Chu et al., 2019; Deng et al., 2020; Gong et al., 2019; Jin et al., 2019; Shan et al., 2019; Su et al., 2019a, 2019b; Wei et al., 2018a, 2018b; Xu et al., 2018a, 2018b, 2018c; Zhang et al., 2019a, 2019b). These metrics are formulated as follows: (7) Sn=TPTP+FN

(8) Sp=TNTN+FP

(9) Acc=TP+TNTP+TN+FP+FN

(10) MCC=TP∗TN−FP∗FN(TP+FP)∗(TP+FN)∗(TN+FP)∗(TN+FN)

TP, TN, FP and FN represent true positive, true negative, false positive, and false negative, respectively. Sn, Sp, Acc, and MCC can be calculated from these indicators. In addition, AUC (area under the ROC curve) was used to evaluate our model (Cheng & Hu, 2018; Cheng et al., 2018b; Ding, Tang & Guo, 2019a, 2019b; Shen et al., 2019). For further experiments, Table 2 records the hyperparameters of the model.

Table 2 The parameters of each experiment.

Experiment	lr	hidden_dim	dropout	Bach_size	
Fig. 2	0.001	100	0.3	64	
Table 3	0.001	100	0.3	64	
Table 4-cv	0.005	100	0.3	64	
Table 4-ind	0.005	100	0.3	64	
Table 5	0.001	100	0.3	64	

Table 3 Performance comparison between 6mA-Pred and other methods via 5-fold cross validation based on the rice dataset.

Method	Sn (%)	Sp (%)	Acc (%)	MCC	AUC	
SNNRice6mA	93.67	86.74	90.20	0.81	0.96	
SNNRice6mA-large	94.33	89.75	92.04	0.84	0.97	
iDNA6mA-rice	93.00	90.50	91.70	0.84	0.96	
6mA-Pred	95.66	92.38	94.02	0.88	0.981	

Table 4 Performance of 6mA-Pred evaluated via 5-fold cross validation and independent testing based on the Mus musculus dataset.

Method	Sn (%)	Sp (%)	Acc (%)	MCC	AUC	
6mA-Pred-cv	93.8	98.5	96.1	0.92	0.981	
6mA-Pred-ind	87.8	98.4	93.8	0.861	0.949	
IDNA6mA-PseKNC	93.28	100	96.73	0.93	–	

Table 5 Performance of 6mA-Pred evaluated via independent testing based on the human dataset.

Method	Sn (%)	Sp (%)	Acc (%)	MCC	AUC	
6mA-Pred-ind	93.28	94.2	93.34	0.87	0.98	
SICD6mA	93.33	95.00	93.66	0.874	–	

Performance comparison with different datasets

Methods for identifying sites in the rice genome include iDNA6mA-Rice and SNNRice6mA, which are excellent models. After comparing different features in feature extraction, the developers of iDNA6mA-Rice chose binary encoding, and they chose RF (random forest) for the classifier. Both the choice of feature method and the performance of the classifier are excellent. iDNA6mA-Rice was applied to various scale segmentation experiments on a rice dataset and achieved very good results. 6mA-Pred was applied in a similar experiment with the rice dataset. the results are shown in Fig. 3. The performance of 6mA-Pred was better than iDNA6mA-Rice at all ratios. However, iDNA6mA-Rice is also a very good model, and the performance difference between the two models was very small. SNNRice6mA also performs very well for rice genes. Unlike iDNA6mA-Rice, SNNRice6mA uses a deep learning model. SNNRice6mA uses one-hot in the feature encoding stage and has achieved good results. Regarding the overall network structure, SNNRice6mA uses a stack structure of CNN (convolutional neural networks). The network structure of SNNRice6mA was adjusted to derive SNNRice6mA-large, which also achieved good results. SNNRice6mA and SNNRice6mA-large were employed for five-fold cross-validation on the rice dataset. Table 3 shows the results of comparisons among the different models. The performance of 6mA-Pred was excellent compared to that of the other models.

Figure 3 Predictive performance at different ratios for the rice dataset.

(A–D) correspond to the performance of the model on different proportions of the rice dataset, respectively.

The model also performed well on the Mus musculus dataset. iDNA6mA-PseKNC has achieved good results in predicting 6mA loci in the Mus musculus genome and uses machine learning methods for analysis. iDNA6mA-PseKNC uses NCP as the feature algorithm, and many experiments have been conducted for this feature. In addition, iDNA6mA-PseKNC employs the SVM classifier and achieved very good results. 6mA-Pred is also effective in identifying 6mA sites in the Mus musculus genome. In this study, two experiments were conducted with 6mA-Pred, one involving five-fold cross-validation on the dataset, and one involving independent testing by splitting the dataset. Table 4 shows the results of these two experiments and the results for iDNA6mA-PseKNC. iDNA6mA-PseKNC was evaluated via the jackknife test; for deep learning methods, leave-one-out cross-validation is time consuming and not representative. For evaluation of 6mA-Pred, five-fold cross-validation (Fang et al., 2019; He et al., 2018a; Liu, 2019; Xiong et al., 2018; Xu et al., 2019b; Zhu et al., 2019) and segmentation of the dataset were employed. As shown in Table 4, the performance of 6mA-Pred remained good.

Among the models used for identifying the 6mA sites of human genes, SICD6mA is currently the best model. SICD6mA is a deep learning model and uses GRU as the basic unit. SICD6mA performs well not only for human genes but also for rice genes. The developers of SICD6mA contributed data and performed extensive data processing. We used the training set and test set provided by SICD6mA’s developers for our experiments. SICD6mA does not use one-hot for encoding; rather, it uses 3-mer. Two basic units, BGRU and UGRU, are used in the network model structure, and a two-layer fully connected layer and a Softmax layer are used to improve the network. The experimental results revealed that the performance of SICD6mA was very good. Table 5 shows the experimental results for 6mA-Pred, which were very similar to the SICD6mA results. These findings proved that 6mA-Pred is very effective in identifying 6mA sites in human genes.

According to the previous conclusions, we conducted related experiments on traditional machine learning methods. NCP and KMER were used in experiments as excellent feature extraction methods. SVM, RF and XGB were excellent algorithms and performed well in previous studies. Therefore, we use them to carry out further experiments. the experimental results are shown in Fig. 4.

Figure 4 Performance comparison between 6mA-Pred and other machine learning methods independent testing based on all datasets.

(A–C) correspond to the performance of commonly used machine learning models under KMER features of different species, respectively. (D–F) are the resulting contrasts under NCP features.

Conclusion

Through the analysis of current studies and the performance of a large number of experimental comparisons, we found that 6mA-Pred is an effective method for identifying 6mA sites. LSTM performs well in processing sequence features and can obtain good features. In addition, the attention mechanism we used is effective for identifying 6mA sites. The combination of LSTM and Attention mechanism can produce a theoretically excellent model, and the experiment proves that this conclusion is correct. Related methods will be considered for RNA and protein modification prediction (Dou et al., 2020; He, Wei & Zou, 2018; Huang & Li, 2018) in the future.

The previous studies on this topic are excellent and provide theoretical and experimental support for our research. The attention mechanism in 6mA-Pred can be improved; for example, self-attention or a combination of two attention mechanisms could be used to obtain a better context vector. It is also possible to use a combination of CNN and attention mechanism to obtain an excellent method (Su et al., 2014). These possibilities warrant investigation.

Additional Information and Declarations

Competing Interests

Author Contributions

Data Availability

The authors declare that they have no competing interests.

Qianfei Huang conceived and designed the experiments, performed the experiments, prepared figures and/or tables, authored or reviewed drafts of the paper, and approved the final draft.

Wenyang Zhou conceived and designed the experiments, prepared figures and/or tables, authored or reviewed drafts of the paper, and approved the final draft.

Fei Guo analyzed the data, prepared figures and/or tables, authored or reviewed drafts of the paper, and approved the final draft.

Lei Xu analyzed the data, authored or reviewed drafts of the paper, and approved the final draft.

Lichao Zhang analyzed the data, authored or reviewed drafts of the paper, and approved the final draft.

The following information was supplied regarding data availability:

Raw data is available at GitHub: https://github.com/huangqianfei0916/6ma-rice.

Code is also available at GitHub: https://github.com/huangqianfei0916/Attention_Classification/tree/master/lstm_attention.

A Web Server For Predicting 6mA Sites is available at:

http://39.100.246.211:5004/6mA_Pred/.

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
