# Peer review of "mA-Pred: identifying DNA N6-methyladenine sites based on deep learning"

_PeerJ, doi:10.7717/peerj.10813_

## Round 0.1 · original submission · Major Revisions

Dear Dr. Huang and colleagues:

Thanks for submitting your manuscript to PeerJ. I have now received three independent reviews of your work, and as you will see, the reviewers raised some concerns about the research. Despite this, these reviewers are optimistic about your work and the potential impact it will lend to research predicting DNA N6-methyladenine sites. Thus, I encourage you to revise your manuscript, accordingly, taking into account all of the concerns raised by the reviewers.

Please consider expanding your work a bit, mostly by providing all of the parameters (and rationale for choosing them) and comparisons to other established approaches. This latter part is crucial for effectively arguing the superiority of 6mA-Pred.

While the concerns of the reviewers are relatively minor, this is a major revision to ensure that the original reviewers have a chance to evaluate your responses to their concerns.

I look forward to seeing your revision, and thanks again for submitting your work to PeerJ.

Good luck with your revision,

-joe

Reviewer 1 ·

Basic reporting

no comment.

Experimental design

no comment.

Validity of the findings

no comment.

Additional comments

In the calculation method of 6mA site recognition, deep learning is rarely applied. The paper applied the attention mechanism of deep learning to 6mA site recognition and achieved good results. The method proposed in the paper has achieved good results in the identification of 6mA site of rice, Mus musculus and human genes. The experimental methods are reasonable and the experimental conclusions are convincing. There are the following suggestions or questions.
1.The description of the model structure is a bit vague. It is recommended to add a detailed description of the model.
2.Why use kmer word segmentation and what are its advantages?
3.Why the dot product method is used when calculating the attention score?
4.What are the advantages of the attention mechanism of the paper compared with the self-attention in Transformer?
5.There are a few misspellings of words in the paper.
Finally, it is recommended to accept this paper.

·

Basic reporting

Most computing methods for 6mA site recognition are still machine learning models。At present, there are relatively few papers recognized by 6mA sites in multiple species。Using a deep learning model to solve the recognition of cross-species 6mA site recognition is indeed relatively novel. From the results, the model performs well in cross-species identification. However, there are still some doubts about the details of the paper.

Experimental design

no comment

Validity of the findings

no comment

Additional comments

① Why doesn't the embedding layer adopt one-hot?
② What are the advantages of combining KMER and embedding?
③ Does the model have any special handling in dealing with cross-species recognition?
④ How about replacing the sequence model with a convolutional neural network?

Reviewer 3 ·

Basic reporting

The manuscript presents a new predictor for DNA 6mA sites. I have several comments for this manuscript that need to be fixed. Please refer to the general comments.

Experimental design

The manuscript presents a new predictor for DNA 6mA sites. I have several comments for this manuscript that need to be fixed. Please refer to the general comments.

Validity of the findings

The manuscript presents a new predictor for DNA 6mA sites. I have several comments for this manuscript that need to be fixed. Please refer to the general comments.

Additional comments

The manuscript presents a new predictor for DNA 6mA sites. I have several comments for this manuscript that need to be fixed.

Major comments:
- For data preprocessing. The authors collected the dataset from different sources, and it seems they do not remove the homology sequences. Approaches such as CD-HIT should be applied with a reasonable threshold to remove the redundancy sequences in this combined dataset.

- The authors should provide a more detailed description of the datasets. How do they select the negative dataset? Is there any overlap in both positive and negative sample between the two datasets they used?

- The authors should clarify why do they use the Kmer instead of One-hot encoding? Did they compare these two encoding schemes on the same condition?

- The authors should provide the experimental results to clarify why they use k=3 in Kmer instead of others.

- The authors should also compare their method with some conventional machine learning algorithms, such as XGBoost, GBDT, SVM, Random Forest, lightGBM and logistic regression

- It would be better to develop an online webserver to facilitate the users to use their method.

- Some loosely associated citations can be further refined.

---

## Round 0.2 · accepted · Accept

Dear Dr. Huang and colleagues:

Thanks for again revising your manuscript. I now believe that your manuscript is suitable for publication. Congratulations! I look forward to seeing this work in print, and I anticipate it being an important resource for research predicting DNA N6-methyladenine sites. Thanks again for choosing PeerJ to publish such important work.

Best,

-joe

Reviewer 1 ·

Basic reporting

Identifying DNA N6-methyladenine sites

Experimental design

Deep Learning was used.

Validity of the findings

Good accuracy was obtianed.

Additional comments

The paper can be accepted.

·

Basic reporting

none

Experimental design

none

Validity of the findings

none

Additional comments

none

Reviewer 3 ·

Basic reporting

The authors did not fully address my comments.

Experimental design

The authors did not fully address my comments.

Validity of the findings

The authors did not fully address my comments.

Additional comments

The authors did not fully address my comments.